# NTS *Prlh* overcomes orexigenic stimuli and ameliorates dietary and genetic forms of obesity

Wenwen Cheng [1✉], Ermelinda Ndoka[1], Jessica N. Maung [2], Warren Pan[1], Alan C. Rupp[1], Christopher J. Rhodes [3], David P. Olson[2,4] & Martin G. Myers Jr [1,2✉]

Calcitonin receptor (*Calcr*)-expressing neurons of the nucleus tractus solitarius (NTS; Calcr[NTS] cells) contribute to the long-term control of food intake and body weight. Here, we show that *Prlh*-expressing NTS (Prlh[NTS]) neurons represent a subset of Calcr[NTS] cells and that *Prlh* expression in these cells restrains body weight gain in the face of high fat diet challenge in mice. To understand the relationship of Prlh[NTS] cells to hypothalamic feeding circuits, we determined the ability of Prlh[NTS]-mediated signals to overcome enforced activation of AgRP neurons. We found that Prlh[NTS] neuron activation and *Prlh* overexpression in Prlh[NTS] cells abrogates AgRP neuron-driven hyperphagia and ameliorates the obesity of mice deficient in melanocortin signaling or leptin. Thus, enhancing *Prlh*-mediated neurotransmission from the NTS dampens hypothalamically-driven hyperphagia and obesity, demonstrating that NTS-mediated signals can override the effects of orexigenic hypothalamic signals on long-term energy balance.

[1] Department of Internal Medicine, University of Michigan, Ann Arbor, MI, USA. [2] Department of Molecular and Integrative Physiology, University of Michigan, Ann Arbor, MI, USA. [3] Research and Early Development, Cardiovascular, Renal, and Metabolism, BioPharmaceuticals R&D, AstraZeneca, Gaithersburg, MD, USA. [4] Division of Endocrinology, Department of Pediatrics, University of Michigan, Ann Arbor, MI, USA. ✉email: wenwenc@umich.edu; mgmyers@umich.edu

The worldwide incidence of obesity (and its comorbid diseases, including diabetes, cardiovascular disease, and many cancers) continues to increase, incurring enormous costs for society in addition to its toll on affected individuals. Unfortunately, existing medical options to prevent and treat obesity remain inadequate to combat the rising tide of this disease. Thus, we must understand mechanisms that contribute to energy balance and that may represent therapeutic targets for obesity[1–3].

Neural and humoral signals from the gut activate satiety circuits in the brainstem *nucleus tractus solitarius* (NTS) to suppress feeding. Some gut-brain axis models suggest that NTS circuits necessarily suppress feeding through aversive responses associated with gut malaise when strongly activated. We found that while some types of NTS neurons (e.g., those that express *Cck*) promote aversive responses with anorexia, others (including distinct sets of NTS neurons that express calcitonin receptor (*Calcr*) or leptin receptor (*Lepr*) (Calcr[NTS] and LepRb[NTS] neurons, respectively)), non-aversively suppress food intake[4,5]. Thus, NTS cell types that mediate aversive signals differ from those that suppress food intake without aversion; systems that mediate the non-aversive suppression of food intake represent more attractive targets for the treatment of obesity and will likely improve patient compliance.

Furthermore, while brainstem circuits were previously thought to control only short-term parameters of food intake (rather than long-term feeding and energy balance), silencing Calcr[NTS] cells increases food intake and weight gain, especially in high fat diet (HFD)-fed animals[4]. Hence, at least some NTS circuits participate in the long-term control of feeding and body weight, and these circuits appear to play a more prominent role in restraining food intake and weight gain during exposure to HFD.

Gut-sensing vagal afferents that innervate the NTS, along with some populations of NTS neurons (including Calcr[NTS] cells), inhibit orexigenic agouti-related protein (AgRP)-containing neurons that reside in the hypothalamic arcuate nucleus (ARC)[4,6,7]. Hence, the modulation of hypothalamic systems that control energy balance may contribute to NTS-mediated feeding suppression. To understand the function of Calcr[NTS] neurons and how they might modify the function of hypothalamic feeding circuits, we examined their gene expression profile, revealing their expression of prolactin releasing hormone (*Prlh*), which encodes the anorectic prolactin releasing peptide (PRRP). While named based upon its ability to increase prolactin release from cell lines in culture, PRRP plays no role in prolactin release in vivo[8]. Instead, *Prlh*, which is primarily expressed in the NTS, the lateral reticular nucleus (LRt) and the dorsomedial hypothalamic nucleus (DMH), modulates food intake and energy expenditure[9–12].

We sought to understand roles and mechanisms of action for *Prlh*-expressing NTS cells (Prlh[NTS] neurons) in the control of food intake and energy balance. We found that that Prlh[NTS] neuron signaling non-aversively abrogates food intake and weight gain, especially during HFD feeding. Furthermore, Prlh[NTS] neurons and NTS *Prlh* overexpression block food intake and promote weight loss during the forced activation of AgRP neurons and in mice with DIO or genetic obesity syndromes. Thus, augmenting specific NTS-mediated satiety signals can override orexigenic hypothalamic signals to suppress feeding and treat obesity, potentially without aversive effects such as nausea.

## Results

### Identification of a *Prlh*-expressing subset of NTS[Calcr] neurons.
We previously demonstrated that Calcr[NTS] neurons mediate the non-aversive suppression of food intake and contribute to the long-term control of food intake and body weight[4]. To understand mechanisms of action for Calcr[NTS] neurons, we used Calcr[eGFP-L10a]

mice that express an eGFP-tagged L10a ribosomal subunit selectively in Calcr neurons (Supplementary Fig. 1A), permitting Calcr neuron−derived ribosomes and their associated mRNA by anti-eGFP translating ribosome affinity purification (TRAP). We subjected TRAP-purified mRNA from hindbrain Calcr neurons to RNA-seq (TRAP-seq) (Supplementary Fig. 1B). Although *Calcr[cre]* incompletely mediates recombination of the *Rosa26[eGFP-L10a]* allele, TRAP material demonstrated the enrichment of *Calcr* (2.8-fold), *Gfp* (2.7-fold), and *Cre* (4.9-fold). This analysis also revealed the enrichment of multiple additional genes (including *Prlh*) in Calcr relative to non-Calcr hindbrain cells (Supplementaryl Fig. 1C).

Consistent with this TRAP-seq analysis, and as previously reported[13–15], PRRP-immunoreactivity (-IR) colocalized with a subset of Calcr[NTS] neurons (Supplementary Fig. 1D). Despite the incomplete recombination of *Rosa26[eGFP-L10a]* mediated by *Calcr[cre]*, the majority of PRRP-IR colocalized with *Calcr*-expressing neurons in the NTS (PRRP + Calcr/PRRP: 61 +/−5%, n = 4); the proportion of Calcr cells colocalizing with PRRP-IR was lower (PRRP + Calcr/Calcr: 46 +/−2%, n = 4), however, suggesting the existence of a substantial population of non PRRP-containing Calcr[NTS] neurons. We also detected PRRP-IR in 62 +/−9% (n = 3) of the few *Calcr*-expressing neurons of the LRt (Supplementary Fig. 1E). We detected no colocalization in the DMH (Supplementary Fig. 1F), however, and no other brain regions contained detectable PRRP-IR. Thus, most *Prlh*-expressing NTS (Prlh[NTS]) neurons represent a subset of Calcr[NTS] cells, suggesting a potentially important role for PRRP neurotransmission by these cells.

To manipulate Prlh[NTS] neurons and determine their function, we generated a *Prlh[Cre]* mouse line (Fig. 1A). Crossing *Prlh[Cre]* onto the cre-inducible *Rosa26[eGFP-L10a]* background (Prlh[eGFP-L10a] mice) revealed GFP-IR in neurons in the expected locations (NTS (Fig. 1B), LRt (Supplementary Fig. 2A, panel a), and DMH (Supplementary Fig. 2A, panel b)), which colocalized with PRRP-IR in the NTS of Prlh[eGFP-L10a] mice (Fig. 1B, C). 100% of PRRP-IR NTS cells (n = 3) overlapped with GFP-IR. Only 74 +/−1% of GFP-IR cells contained PRRP-IR, however, and we speculate that the failure of PRRP-IR to detect all Prlh[NTS] neurons may underlie the detection of *Prlh[cre]*-expressing neurons that do not contain detectable PRRP. Thus, *Prlh[Cre]* mediates cre-dependent recombination in *Prlh* neurons.

We examined projections from Prlh[NTS] cells by injecting AdV-iN-Syn-mCherry (which mediates the cre-dependent expression of the synaptically-targeted synaptophysin-mCherry (Syn-mCherry) fusion protein) into the NTS of *Prlh[Cre]* mice. As previously described for Calcr[NTS] cells[4], this analysis revealed projections from Prlh[NTS] cells to the parabrachial nucleus (PBN), the paraventricular hypothalamic nucleus (PVH), the DMH, and the bed nucleus of the stria terminalis (BNST) (Supplementary Fig. 2B). We also employed cre-dependent rabies-mediated single-synapse retrograde tracing[4,16] to define the neurons synapsing on Prlh[NTS] cells. Regions containing substantial numbers of neurons presynaptic to Prlh[NTS] include the lateral hypothalamic area (LHA), the PVH, and the amygdala (Supplementary Fig. 2C). Thus, Prlh[NTS] cells possess similar upstream and downstream connections as the larger Calcr[NTS] population[4].

### Roles for Prlh[NTS] neurons and NTS *Prlh* in the control of food intake and body weight.
To understand the function of Prlh[NTS] neurons, we injected control AAV or AAV[Flex-hM3Dq] (which mediates the cre-dependent expression of the activating (hM3Dq) isoform of the Designer Receptors Exclusively Activated by Designer Drugs (DREADD)) into the NTS of *Prlh[Cre]* mice. As expected, administration of the DREADD activator, clozapine-N-oxide (CNO), to these Prlh[NTS-Dq] animals promoted FOS-IR accumulation in Prlh[NTS] neurons (Fig. 1D, E), consistent with the

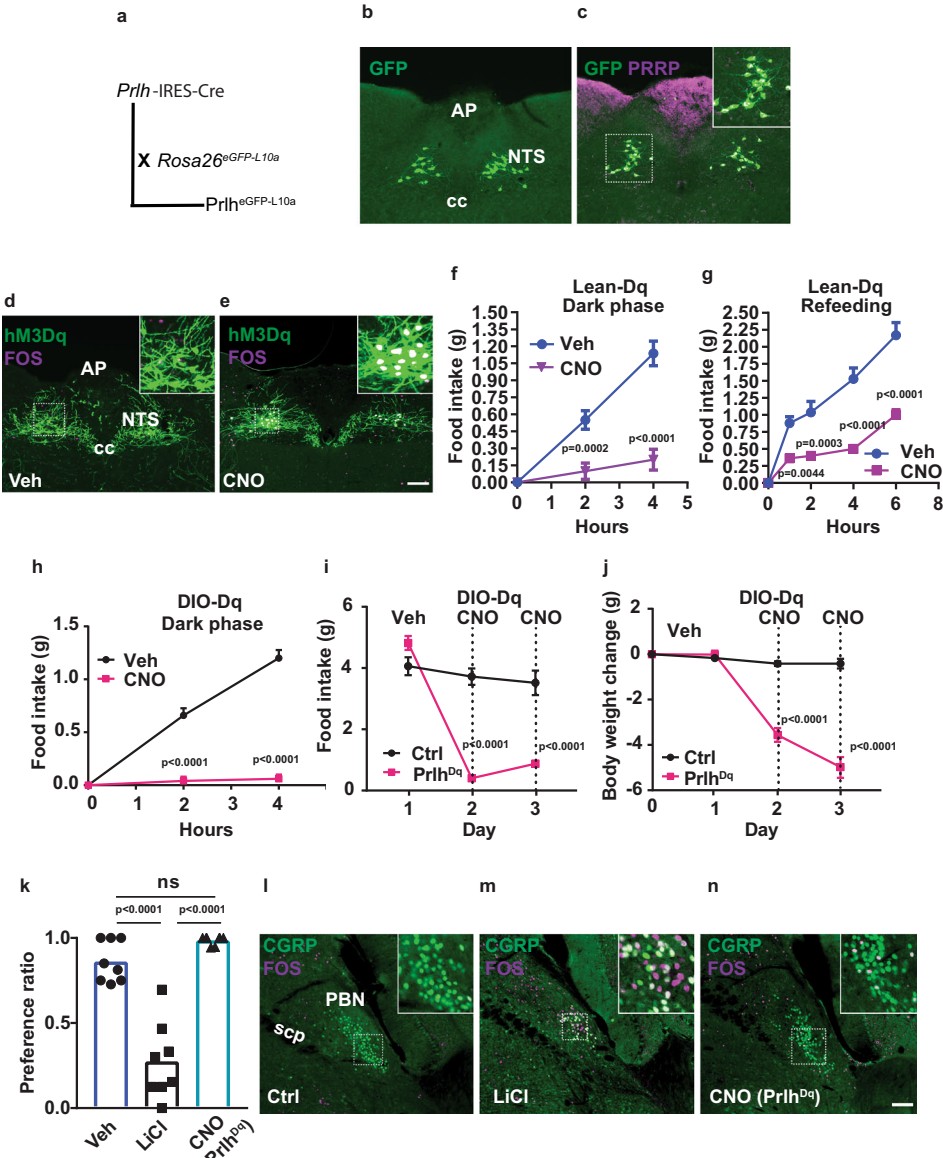

**Fig. 1 $Prlh^{cre}$ mice reveal roles for $Prlh^{NTS}$ neurons in the control of food intake in lean and DIO mice. A** Schematic diagram showing the $Prlh^{Cre}$ mouse strain and its breeding onto the $ROSA26^{eGFP-L10a}$ background to generate $Prlh^{eGFP-L10a}$ reporter mice. **B, C** Representative images showing GFP-IR (green) alone (**B**) or together with PRRP-IR (magenta) (**C**) in the NTS of a $Prlh^{eGFP-L10a}$ reporter mouse. Inset in (**C**) shows digitally zoomed images of the boxed region. **D, E** Representative images showing mCherry-IR (hM3Dq, green) and FOS-IR (magenta) in the NTS of $Prlh^{NTS-Dq}$ mice following treatment with saline (**D**, Veh) or CNO (**E**, IP, 1 mg/kg) for 2 h before perfusion. **F, G** We examined food intake in chow-fed $Prlh^{NTS-Dq}$ mice over the first 4 h of the dark phase (**F**, $n = 8$ animals/group) and during the first 6 h of refeeding in the light cycle following an overnight fast (**G**, $n = 7$ animals/group) during treatment with CNO (IP, 1 mg/kg) or Veh. **H** DIO $Prlh^{NTS-Dq}$ mice were treated with Veh or CNO (IP, 1 mg/kg) immediately prior to the onset of the dark cycle and food intake was measured over the subsequent four hours. **I, J** DIO Control (Ctrl, $n = 5$ animals/group) or $Prlh^{NTS-Dq}$ ($Prlh^{Dq}$, $n = 6$ animals/group) mice were treated with Veh for two baseline days, followed by two days with CNO (a single IP 1 mg/kg dose at the onset of the dark cycle followed by 3.33 mg/ml in drinking water for the duration of treatment) and daily food intake (**I**) and body weight change from baseline (**J**) were determined. Vehicle (Veh) and CNO treatment are denoted on the graphs. **K** Mice were treated with vehicle (Veh), LiCl, or CNO (IP) during exposure to a novel tastant (HFD); their consumption of HFD given a choice between chow and HFD was determined the following day; $n = 8$ animals/group. **L–N** Representative images showing PBN FOS-IR (purple) and GFP-IR (CGRP, green) in control mice (Ctrl), mice treated with LiCl (IP, 126 mg/kg), or in $Prlh^{NTS-Dq}$ mice treated with CNO (IP, 1 mg/kg); all mice were on the $CGRP^{GFP}$ background[20]. Shown is mean +/− SEM. Two-way ANOVA, sidak's multiple comparisons test was used; $p$ values are shown for significant comparisons. All images taken at same magnification; scale bar equals 150 µm. ns: not significant vs Veh ($p > 0.05$). All experiments were repeated in two independent cohorts of animals with similar results; cohorts were combined for publication.

hM3Dq-mediated activation of these cells. As for $Calcr^{NTS}$ cells[4], activation of $Prlh^{NTS}$ cells suppressed food intake in a variety of acute paradigms (Fig. 1F, G). The hM3Dq-mediated activation of $Prlh^{NTS}$ cells also suppressed food intake and decreased body weight over multiple days in diet-induced obese (DIO) mice (Fig. 1H–J), consistent with the potential utility of $Prlh^{NTS}$

circuits as targets for the treatment of obesity. Furthermore, activating $Prlh^{NTS}$ neurons did not provoke a conditioned taste aversion (CTA, Fig. 1K), and poorly promoted FOS-IR in aversive $CGRP^{PBN}$ neurons (FOS + CGRP/CGRP: Veh- 2 +/−0.3%; LiCl- 37 +/−9% ($p = 0.0115$ vs Veh); CNO- 9 +/−4% ($p = 0.5925$ vs Veh), $n = 3$ animals/group) (Fig. 1L–O).

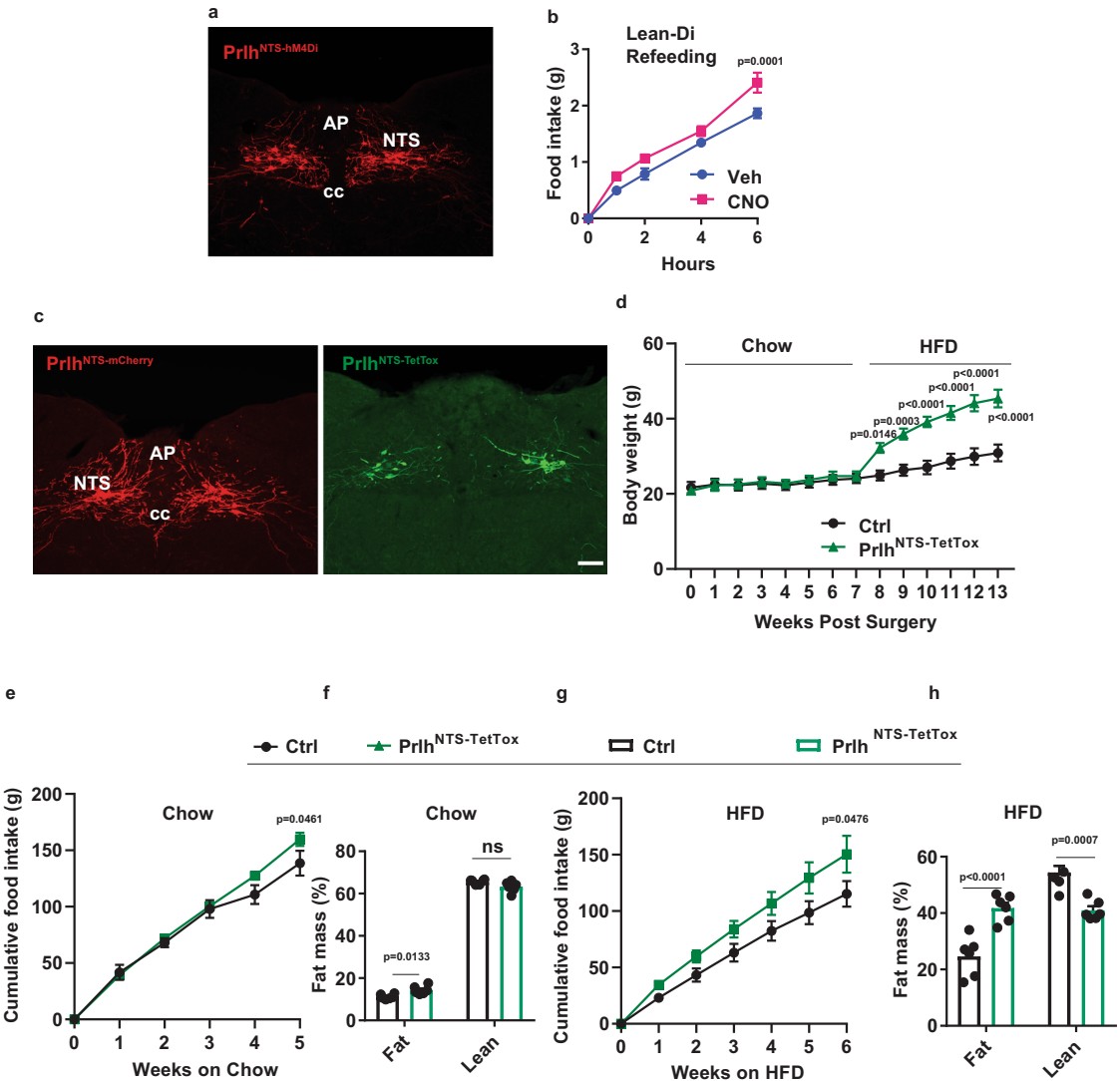

**Fig. 2 Silencing Prlh^NTS neurons promotes DIO. A** Representative NTS image showing dsRed-IR (red, mCherry) from *Prlh^cre* mice injected with mCherry-expressing AAV^Flex-hM4Di (Prlh^NTS-hM4Di) in NTS. **B** Food intake during the first 6 h following an overnight fast in Prlh^NTS-hM4Di mice during treatment with vehicle (Veh) or CNO injection (IP, 1 mg/kg), *n* = 8 in each group. **C** Representative NTS images showing dsRed-IR (red, left panel) and GFP-IR (green, right panel) from *Prlh^cre* mice injected with mCherry-expressing control AAV (Prlh^NTS-mCherry; Ctrl, left panel) or GFP-expressing AAV^Flex-TetTox (Prlh^NTS-TetTox, right panel). **D** Body weight of control (Ctrl) and Prlh^NTS-TetTox mice following surgery, during which time they were fed with chow for 7 weeks and HFD for an additional 6 weeks. **E–H** Food intake (**E**, **G**) and body composition (**F**, **H**) at the end of the 7th week (**F**) and 13th week (**H**) after surgery for Ctrl and Prlh^NTS-TetTox mice during Chow (**E**, **F**, *n* = 6 animals/group) and HFD (**G**, **H**, *n* = 4 Ctrl animals and *n* = 6 Prlh^NTS-TetTox animals) feeding. Shown is mean +/− SEM. Two-way ANOVA, sidak's multiple comparisons test was used: *p* values are shown for significant comparisons. All images were taken at the same magnification; scale bar equals 150 μm. All experiments were repeated in two independent cohorts of animals with similar results; cohorts were combined for publication.

To understand physiological roles for Prlh^NTS neurons, we inactivated or silenced them and examined the resultant effects on food intake and body weight. Following the bilateral injection of AAV^Flex-hM4Di into the NTS of *Prlh^Cre* mice to cre-dependently express the inhibitory (hM4Di) DREADD in Prlh^NTS cells, CNO increased food intake during refeeding following an overnight fast (Fig. 2A, B). We also injected AAV^Flex-TetTox bilaterally into the NTS of *Prlh^cre* mice to express tetanus toxin (TetTox) in and silence Prlh^NTS neurons (Prlh^NTS-TetTox mice) (Fig. 2C). Chow-fed Prlh^NTS-TetTox mice tended to display increased body weight (although not significantly) and exhibited small increases in overall food intake and adiposity (which were observed only at late times due to decreases in food intake by control animals) (Fig. 2E–F). Silencing Prlh^NTS cells dramatically increased food intake, body weight, and adiposity over 6 weeks in Prlh^NTS-TetTox mice exposed

to HFD, however (Fig. 2D, G, H). Note that these mice are on a segregating background and are not as sensitive to DIO as are pure C57 mice; inactivation of NTS *Prlh* increases this sensitivity to DIO, however. Thus, like the larger population of Calcr^NTS cells, Prlh^NTS neurons non-aversively suppress feeding and participate in the long-term regulation of food intake and body weight, especially in HFD-fed mice.

Because *Prlh* and the PRRP receptor (a.k.a., GPR10, encoded by *Prlhr*) both contribute to energy balance[9–12], we sought to determine the potential physiological role for NTS *Prlh* expression in the control of energy balance. We thus generated a Cre-conditional *Prlh^Flox* mouse allele and crossed it onto the *Calcr^cre* background to produce mice lacking *Prlh* in Calcr cells, including those in the NTS (Prlh^CalcrKO mice) (Fig. 3A). Immunostaining confirmed the depletion of ~90% of PRRP-IR soma from the NTS

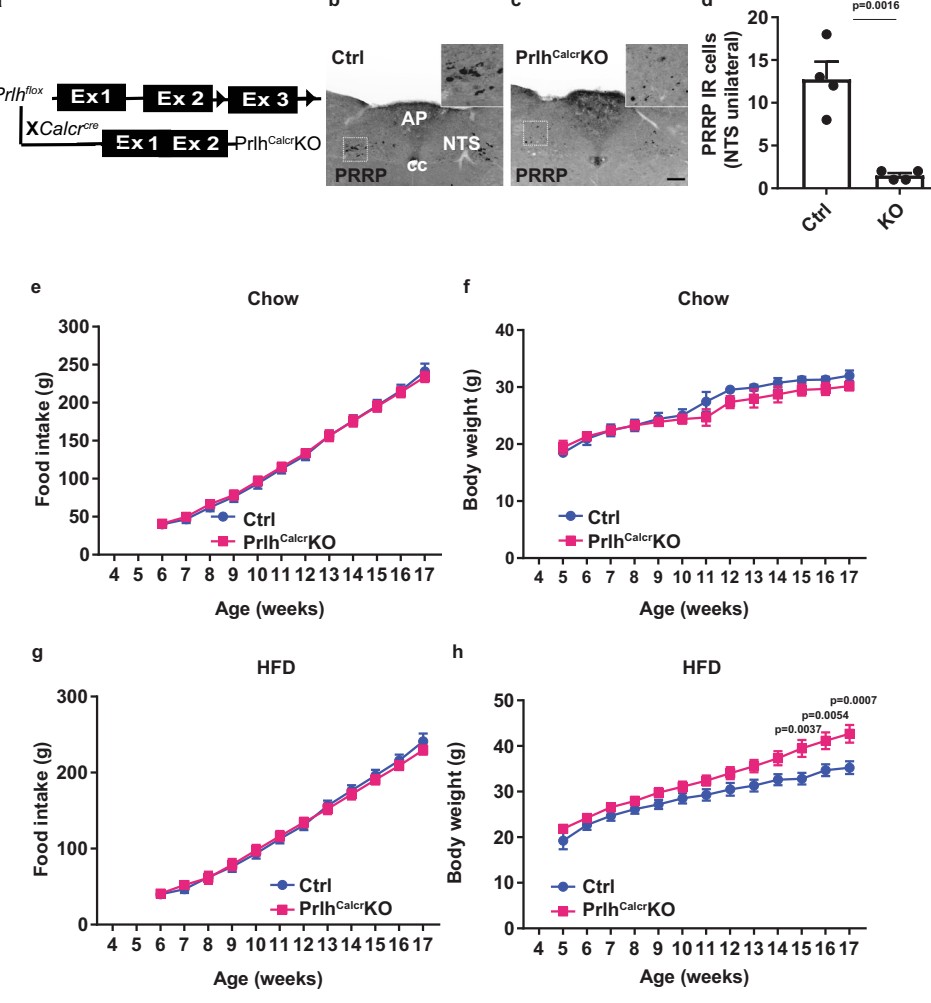

**Fig. 3 Deletion of *Prlh* from CalcrNTS cells exacerbates DIO. A** Schematic diagram showing the cross of *PrlhFlox* onto the *Calcrcre* background to generate *Calcrcrecre*; *Prlhflox/flox* (PrlhCalcrKO) mice. **B**, **C** Representative images showing PRRP-IR in the NTS of *Calcrcre/cre*; *Prlh+/+* control (**B**, Ctrl) and PrlhCalcrKO (**C**) mice. Insets show digital zooms of the boxed regions. All images taken at same magnification; scale bar equals 150 μm. **D** Quantification of PRRP-IR in the NTS of control (Ctrl) and PrlhCalcrKO mice, as in B, C; *n* = 4 animals/group. **E**–**H** Weekly food intake (**E**, **G**) and body weight (**F**, **H**) for Ctrl and PrlhCalcrKO fed with chow (**E**, *n* = 7 animals/group; **F**, *n* = 6 Ctrl animals and *n* = 10 PrlhCalcrKO animals) or HFD (**G**, **H**) from the time of weaning at 4 weeks of age. Shown is mean +/− SEM. Two-way ANOVA, sidak's multiple comparisons test was used: *p* values are shown for significant comparisons. All experiments were repeated in two independent cohorts of animals with similar results; were generally combined for publication.

in PrlhCalcrKO mice (Fig. 3B–D); this also suggests that ~90% of PrlhNTS cells express *Calcrcre*. While we found no difference in food intake or body weight between chow-fed control and PrlhCalcrKO mice (Fig. 3E, F), the abrogation of NTS *Prlh* expression increased body weight in older HFD-fed PrlhCalcrKO mice (Fig. 3G, H). We were not able to detect differences in food intake for these animals, however, presumably due to the error inherent in food intake measurements and the small change in food intake that likely underlies the modest increase in body weight in these animals. These data suggest a physiologic role for CalcrNTS neuron *Prlh* expression in the restraint of DIO.

To determine the requirement for *Prlh* in the suppression of food intake and body weight during the artificial activation of CalcrNTS cells, we injected AAVFlex-hM3Dq into the NTS of *Calcrcre* or PrlhCalcrKO mice. In this experimental paradigm, CNO activates the entire CalcrNTS population (rather than just PrlhNTS cells). *PrlhCre* and *PrlhFlox* share the same genetic locus, however, and thus cannot be combined to permit the manipulation of PrlhNTS cells that lack *Prlh*.

We found that the lack of NTS *Prlh* expression failed to alter the suppression of food intake and body weight by the hM3Dq-

mediated activation of CalcrNTS cells (Supplementary Fig. 3), suggesting that non-PRRP neurotransmission must mediate food intake suppression during the artificial activation of the CalcrNTS cells. Indeed, PrlhNTS cells contain glutamatergic markers (Supplementary Fig. 4A, B), and the ablation of *Slc17a6* (which encodes the vesicular glutamate transporter, vGLUT2) from *Prlh*-expressing cells abrogated the suppression of food intake and body weight during hM3Dq-mediated PrlhNTS neuron activation (Supplementary Fig. 4C–I). Interestingly, however, vGlut2 deficiency in PrlhNTS neurons did not alter long-term energy balance in chow- or HFD-fed mice (Supplementary Fig. 4J–M). Hence, non-*Prlh*-dependent glutamate signaling mediates the suppression of food intake during the artificial activation of CalcrNTS cells, even though *Prlh* expression in CalcrNTS neurons contributes to the physiological control of energy balance.

Because the artificial activation of NTS cells does not directly assay physiologic roles for NTS *Prlh*, we sought to increase *Prlh* expression in PrlhNTS cells as a means of amplifying *Prlh*-dependent signaling by these cells. We therefore generated AAVFlex-Prlh to cre-dependently overexpress *Prlh* (Supplementary Fig. 5A). Because anorectic PRRP activity requires its RF-

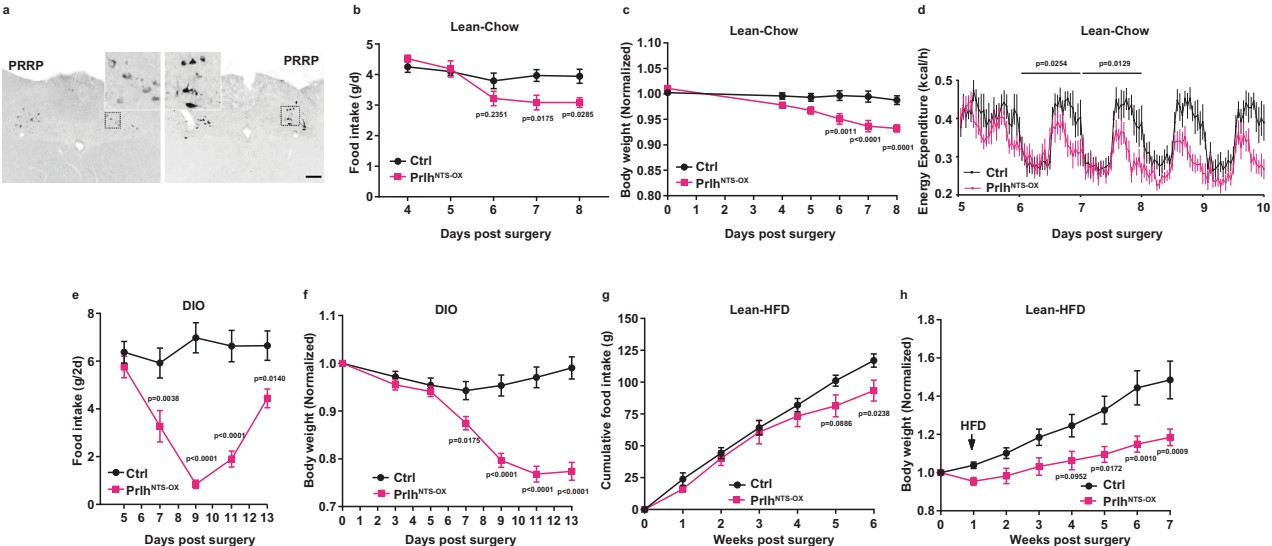

**Fig. 4 Increased *Prlh* expression in Prlh^NTS neurons promotes negative energy balance. A** Representative images of PRRP-IR (black) in the NTS of control (Ctrl, left panel) or Prlh^NTS-OX mice (right panel). All images were captured at the same magnification; scale bar equals 150 µm. Insets show digital zooms of the boxed regions. **B, C** Food intake (**B**) and body weight (**C**) for lean, chow-fed control (Ctrl) and Prlh^NTS-OX mice following surgery (n = 5 Ctrl animals and n = 6 Prlh^NTS-OX animals); body weight is shown normalized to baseline (pre-surgery) weight. **D** Energy expenditure for chow-fed control (Ctrl) and Prlh^NTS-OX mice was determined in metabolic cages for days 5–9 following surgery (n = 8 per group). **E–H** Food intake (**E, G**) and body weight (**F, H**) for control (Ctrl) and Prlh^NTS-OX mice following surgery in (**E, F**) DIO (n = 6 Ctrl animals and n = 7 Prlh^NTS-OX animals), and (**G**, n = 8 Ctrl animals and n = 5 Prlh^NTS-OX animals, **H**, n = 10 animals/group) chow-fed mice switched to HFD 7 days post-surgery (n = 5–10 per group). Body weight is shown normalized to baseline (pre-surgery) weight. Shown is mean +/− SEM. Two-way ANOVA, sidak's multiple comparisons test was used; p values are shown for significant comparisons. All experiments were repeated in two independent cohorts of animals with similar results; cohorts were combined for publication.

amidation[17], we did not fuse a tag to the *Prlh* coding region in AAV^Flex-Prlh, and initially tested the virus by injecting it into the NTS of *Lepr^cre* mice (LepRb^NTS neurons do not contain endogenous PRRP) (Supplementary Fig. 5B). This analysis revealed that AAV^Flex-Prlh mediated the cre-dependent expression of PRRP in LepRb^NTS cells. Furthermore, the injection of AAV^Flex-Prlh into the NTS of *Plrh^Cre* animals (Prlh^NTS-OX mice) increased the intensity of NTS PRRP-IR compared to control mice (Fig. 4A). Thus, AAV^Flex-Prlh promotes PRRP accumulation in cre-expressing neurons and appears to increase the cellular content of PRRP in Prlh^NTS cells.

We found that chow-fed Prlh^NTS-OX mice exhibited decreased food intake and body weight during the first few days after surgery (Fig. 4B, C), as well as demonstrating decreased refeeding following a fast (Supplementary Fig. 5C). Analysis of chow-fed mice in metabolic cages during days 5–9 following surgery revealed decreased energy expenditure in Prlh^NTS-OX mice compared to controls, the magnitude of this effect increased with time after surgery; these data are consistent with an appropriate suppression of energy expenditure due to decreased feeding and body weight in these animals (Fig. 4D). The modestly decreased food intake and body weight in chow-fed Prlh^NTS-OX mice attenuated with time, however (Supplementary Fig. 5D, E). Interestingly, we observed an enhanced and prolonged suppression of food intake and body weight in DIO Prlh^NTS-OX mice (Fig. 4E, F), and found that the overexpression of *Prlh* in Prlh^NTS neurons provided long-term protection from obesity in lean mice exposed to HFD following surgery (Fig. 4G, H). We observed increased FOS-IR in the NTS and PBN of Prlh^NTS-OX mice (Supplementary Fig. 5F, G), suggesting that augmented *Prlh* expression in Prlh^NTS cells increased the activity of downstream neural circuits that may suppress feeding. Hence, increased NTS *Prlh* expression and signaling attenuates the augmentation of food intake and the accretion of body weight that normally results from HFD exposure.

**Prlh^NTS neurons and *Prlh* suppress AgRP neuron-stimulated feeding.** We previously showed that Calcr^NTS cells inhibit the activity of orexigenic AgRP neurons[4]. To determine the potential role for Prlh^NTS neurons in this effect, we examined the ability of CNO-dependent Prlh^NTS neuron activation in Prlh^NTS-Dq mice to suppress fasting-induced mediobasal ARC (mbARC) FOS-IR (Supplementary Fig. 6A–G). Because the activation of Prlh^NTS cells decreased this surrogate for AgRP neuron activity, we tested the role for AgRP neuron inhibition in the suppression of food intake by determining whether the enforced activation of AgRP neurons could block food intake suppression by NTS^Prlh neurons (Fig. 5).

We injected AAV^Flex-hM3Dq into the NTS and AAV^Flex-ChR2 into the ARC of compound *Prlh^Cre;Agrp^Cre* mice (Fig. 5A). Blue light delivery by an ARC-implanted optical fiber in these Prlh^NTS-Dq; Agrp^ChR2 mice stimulates channelrhodopsin (ChR2) in AgRP neurons, enforcing their activation (Fig. 5B), while CNO promotes Prlh^NTS neuron activation, allowing us to assess the aggregate effects of these stimuli on food intake. As expected, the optogenetic activation of AgRP cells in Prlh^NTS-Dq;Agrp^ChR2 mice during the light cycle promoted food intake (Fig. 5C). We also found that activating Prlh^NTS neurons prior to the stimulation of AgRP neurons not only abrogated baseline food intake, but also inhibited the augmentation of feeding by AgRP neuron activation (Fig. 5C). Furthermore, activating Prlh^NTS neurons following an hour of AgRP neuron stimulation blocked subsequent food intake (Fig. 5C). Thus, not only does Prlh^NTS neuron-mediated appetite suppression not require AgRP neuron inhibition, but also Prlh^NTS neuron stimulation blocks the ability of enforced AgRP neuron activation to promote feeding.

To determine whether endogenous PRRP in Prlh^NTS neurons might similarly suppress AgRP neuron-dependent food intake, we injected AAV^Flex-hM3Dq into the ARC of chow-fed *Prlh^Cre;Agrp^Cre* mice followed by the injection of AAV^Flex-Prlh

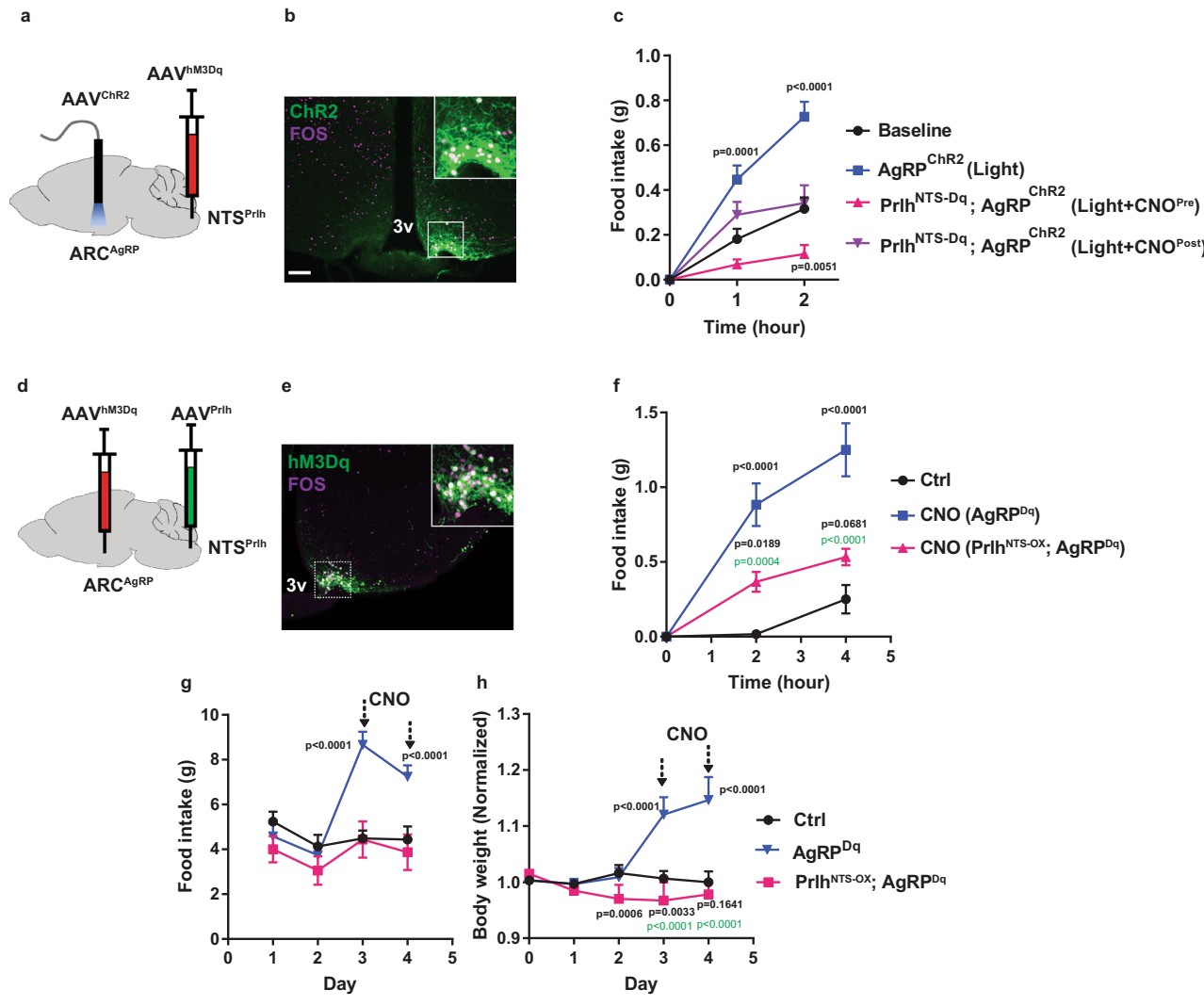

**Fig. 5 $Prlh^{NTS}$ action abrogates AgRP neuron-promoted food intake and weight gain. A** Schematic showing the stereotaxic injection of $AAV^{Flex-hM3Dq}$ into the NTS and $AAV^{Flex-ChR2}$ plus optical fiber into the ARC of $Prlh^{cre};Agrp^{cre}$ mice. **B** Representative image of GFP-IR (ChR2, green) and FOS-IR (magenta) in the ARC of mice treated as in (**A**), following photostimulation for 1 h. **C** Cumulative food intake for the first 1 or 2 h of treatment for mice as in (**A**) that were unstimulated (black), photostimulated (blue), CNO-treated immediately prior to photostimulation ($CNO^{Pre}$, red), and CNO-treated following the first hour of photostimulation ($CNO^{Post}$, purple); $n = 9$ animals in baseline group; $n = 10$ in light group; $n = 7$ in Light + $CNO^{Pre}$ group, and $n = 8$ in Light + $CNO^{Post}$ group. **D** Schematic showing the stereotaxic injection of $AAV^{Flex-hM3Dq}$ into the ARC of $Prlh^{cre};Agrp^{cre}$ mice ($AgRP^{Dq}$ mice); some mice also received $AAV^{Flex-Prlh}$ into the NTS ($NTS^{Prlh};AgRP^{Dq}$ mice). **E** Representative image showing mCherry-IR (hM3Dq, green (pseudocolored)) and FOS-IR (magenta) in the ARC of $NTS^{Prlh};AgRP^{Dq}$ mice following CNO treatment for 2 h. **F** Food intake during the first 4 h of the dark cycle for CNO (IP, 1 mg/kg)-treated mice of the designated experimental groups; $n = 6$ per group. **G–H** Daily food intake (**G**) and body weight (**H**) (measured during the light cycle) for CNO (IP, 1 mg/kg)-treated mice of the designated experimental groups; $n = 6$ animals/group for Ctrl and $NTS^{Prlh};AgRP^{Dq}$; $n = 5$ animals in $AgRP^{Dq}$ group. All graphs: shown is mean $+/-$ SEM. Two-way ANOVA, sidak's multiple comparisons test was used. Significant or near significant $p$ values for comparisons to $AgRP^{ChR2}$ (**C**) or $AgRP^{Dq}$ (**F**), and Ctrl group (**G–H**) are shown in black; those for comparisons between $AgRP^{Dq}$ and $Prlh^{NTS-OX};AgRP^{Dq}$ groups are shown in green. All experiments were repeated in two independent cohorts of animals with similar results; cohorts were combined for publication. All images taken at the same magnification; scale bar equals 150 µm. 3 v = third cerebral ventricle.

into the NTS (Fig. 5D), enabling us to examine the effect of CNO-dependent AgRP neuron activation on feeding in the presence of *Prlh* overexpression in $Prlh^{NTS}$ cells in these $Prlh^{NTS-OX};Agrp^{Dq}$ mice. CNO treatment in $Agrp^{Dq}$ mice activated AgRP neurons (Fig. 5E) and rapidly and sustainably increased feeding over 48 h, promoting increased body weight (Fig. 5F–H). *Prlh* overexpression in $Prlh^{NTS}$ neurons attenuated AgRP neuron-driven acute feeding responses in $Prlh^{NTS-OX};Agrp^{Dq}$ mice (Fig. 5F), despite ongoing activation of AgRP neurons (as assessed by FOS-IR) (Supplementary Fig. 6H–K). Furthermore, $Prlh^{NTS-OX};Agrp^{Dq}$ mice exhibited normalized or decreased food intake and body weight during prolonged (48-h) hM3Dq-mediated AgRP neuron

stimulation (Fig. 5G, H). Thus, the activation of $Prlh^{NTS}$ cells and the overexpression of *Prlh* to increase PRRP content in $Prlh^{NTS}$ cells block AgRP neuron-dependent feeding.

**$Prlh^{NTS}$ neurons and *Prlh* suppress hypothalamic obesity.** The ability of $Prlh^{NTS}$ neuron activation or *Plrh* overexpression to inhibit AgRP neuron-driven food intake and body weight accretion (as well as to their ability to block HFD-provoked weight gain) suggests the potential for the NTS Prlh system to suppress hypothalamic obesity more broadly. To test this notion, we bred $Prlh^{Cre}$ onto the leptin-deficient $Lep^{ob/ob}$ (*ob/ob*)

background or the Agouti ($A^{y/+}$; $A^y$) background. Leptin deficiency increases the activity of AgRP neurons and decreases the activity of anorexigenic proopiomelanocortin (POMC) neurons in the ARC (in addition to causing other defects), resulting in dramatic hyperphagia and obesity[18]. The ubiquitous overexpression of agouti signaling protein in $A^y$ mice does not alter the activity of AgRP and POMC neurons, but rather blocks melanocortin action on the downstream neurons that would otherwise be activated by POMC-derived peptides and inhibited by AgRP; thus, $A^y$ mice exhibit hyperphagia and obesity[19]. We injected AAV$^{Flex-hM3Dq}$ into the NTS of Prlh$^{Cre}$; ob/ob (Prlh$^{NTS-Dq}$;ob/ob mice), Prlh$^{Cre}$; $A^{y/+}$ (Prlh$^{NTS-Dq}$; $A^y$ mice) and Prlh$^{Cre}$ control mice and examined the effects of hM3Dq-mediated Prlh$^{NTS}$ neuron activation in these animals (Fig. 6A–F). We found that the activation of Prlh$^{NTS}$ cells not only suppressed acute food intake in Prlh$^{NTS-Dq}$;ob/ob and Prlh$^{NTS-Dq}$; $A^y$ mice mice during the first 4 h of the dark cycle (Fig. 6A, D), but also sustainably decreased their daily food intake and body weight over two days of treatment (Fig. 6B–C, E–F).

To determine the potential ability of increased PRRP signaling by Prlh$^{NTS}$ cells to mitigate food intake and body weight in models of hypothalamic dysfunction, we again used the $A^y$ and ob/ob models. We injected AAV$^{Flex-Prlh}$ into the NTS of lean control Prlh$^{Cre}$, obese Prlh$^{Cre}$;Lep$^{ob/ob}$ and obese Prlh$^{Cre}$;$A^{y/+}$ mice and monitored subsequent food intake and body weight (Fig. 6G-J). In both models, Prlh overexpression in Prlh$^{NTS}$ cells decreased food intake to near or below control values (Fig. 6G, I). Furthermore, this intervention normalized the trajectory of body weight gain in ob/ob mice and promoted weight loss in $A^y$ mice (Fig. 6H, J). Thus, augmenting Prlh-dependent NTS signals not only reduces food intake and body weight in HFD-fed normal animals, but also abrogates hypothalamus-driven feeding during the artificial activation of AgRP neurons and in genetic models with severe dysregulation of hypothalamic feeding systems.

## Discussion

Seeking to understand potential mechanisms for the non-aversive suppression of food intake and control of long-term energy balance by Calcr$^{NTS}$ neurons, we set out to understand roles for Prlh expression in their function, showing that Prlh$^{NTS}$ cells suppress food intake and body weight (especially in HFD-fed mice) and that NTS Prlh restrains weight gain during HFD exposure (Fig. 7). The inhibition of AgRP neurons is not required for the NTS Prlh-dependent suppression of food intake, but rather increased Prlh-dependent NTS signaling abrogates food intake during the enforced activation of AgRP neurons. Furthermore, amplifying signals from Prlh$^{NTS}$ neurons attenuates food intake and obesity in mice lacking leptin or with attenuated melanocortin signaling. Thus, enhancing Prlh-mediated neurotransmission by the NTS blocks the hyperphagia and obesity associated with leptin deficiency, increased AgRP neuron activity and/or impaired melanocortin signaling, as well as during DIO. These findings demonstrate the ability of Prlh/PRRP specifically, and the NTS in general, to override the effects of orexigenic hypothalamic signals and decrease long-term food intake and body weight.

The manipulation of brainstem satiety systems has generally impacted the size of individual meals, rather than total caloric intake over the long term[20,21]. Calcr$^{NTS}$ and Prlh$^{NTS}$ cells receive feeding-related input from the gut[4,5,14,22] and inhibit feeding when activated; they thus presumably represent part of the NTS satiety circuitry. Because interfering with the function of these cells increases total caloric intake and body weight over the long term, these cells must contribute to the tonic suppression of food intake sufficiently to alter overall food intake. Hence, brainstem satiety systems impact overall energy balance. Furthermore, the

artificial amplification of satiety signals (e.g., by treatment with GLP1R or CALCR agonists) also decreases long-term food intake and body weight, despite the failure of Glp1r or Calcr ablation (or interference with endogenous NTS GLP1 signaling) to alter these parameters[4,23–25]. Hence, the pharmacologic activation of some peptide/receptor-mediated brainstem satiety signals can decrease long-term feeding and body weight, even if their endogenous activity is insufficient to affect these parameters.

Interestingly, endogenous as well as artificially amplified Prlh-dependent signaling by PlrhNTS cells restrains weight gain and promotes weight loss over the long term in HFD-fed mice, even though Prlh-dependent effects on food intake and body weight are small and/or short-lived in chow-fed animals. Consistently, re-expression of Prlh in the NTS decreases food intake and weight gain on HFD relative to Prlh-null animals (DMH Prlh increases energy expenditure)[22]. Hence, PRRP signaling by Prlh$^{NTS}$ cells may play a more important role in preventing the overconsumption of calories than in inhibiting caloric intake appropriate to weight maintenance in lean mice.

We also found that NTS Prlh expression contributes to the long-term modulation of physiologic food intake, but not the suppression of feeding by the DREADD-mediated activation of Calcr$^{NTS}$ cells. While the lack of Prlh-dependent effects on DREADD-mediated feeding suppression in this study might reflect differences between the larger group of Calcr$^{NTS}$ cells and Prlh$^{NTS}$ neurons, we found that glutamatergic signaling by Prlh$^{NTS}$ cells is required for DREADD-mediated food intake suppression by these neurons. Thus, although glutamate signaling by Prlh$^{NTS}$ cells is required for the suppression of food intake and body weight over the short term during the artificial activation of these cells, PRRP in Prlh$^{NTS}$ cells modulates energy balance over the long term. Because ablating NTS Prlh resulted in milder obesity than did silencing Prlh$^{NTS}$ neurons, some non-PRRP-mediated signal from Prlh$^{NTS}$ cells presumably also contributes to energy balance, however. It is possible that the remaining 10–15% of NTS Prlh expression and/or glutamate release from Prlh$^{NTS}$ cells contributes to energy balance in the absence of most Prlh expression by these cells.

Because the activation of gut→vagus→NTS circuits inhibits the firing of orexigenic AgRP neurons[6,7], we speculated that the inhibition of AgRP neurons might contribute to food intake suppression by Prlh$^{NTS}$ cells. We thus examined the ability of Prlh$^{NTS}$ cells to suppress food intake during the enforced firing of AgRP neurons. Not only did the artificial activation of AgRP neurons fail to abrogate Prlh$^{NTS}$-mediated food intake suppression, but also Prlh$^{NTS}$ signaling blocked AgRP neuron-mediated feeding (as well as attenuating hyperphagia and weight gain in models of hypothalamic obesity). Thus, Prlh$^{NTS}$ neuron-mediated signaling must suppress food intake via a mechanism(s) distinct from the inhibition of AgRP neurons. Future research will be required to identify the neural circuits and mechanisms that mediate these effects.

Our anterograde tracing revealed that NTS$^{Prlh}$ neurons innervate several brain regions that also receive direct input from AgRP and/or POMC neurons, including the PBN, BNST and PVH; these regions might represent important points of functional convergence between Prlh$^{NTS}$ cells and hypothalamic systems. Indeed, Prlh overexpression in Prlh$^{NTS}$ cells increases FOS-IR in the PBN, suggesting the increased activation of PBN cells by augmented PRRP release from the NTS. Prlh$^{NTS}$ neuron stimulation poorly activates aversive[20,26,27] CGRP$^{PBN}$ cells and, consistently, fails to promote a CTA. Other PBN targets might contribute to Prlh$^{NTS}$-dependent food intake suppression, however.

Because Prlh/PRRP-dependent signaling downstream of Prlh$^{NTS}$ neurons mediates the suppression of feeding during DIO and other forms of hyperphagic obesity, it will be important to

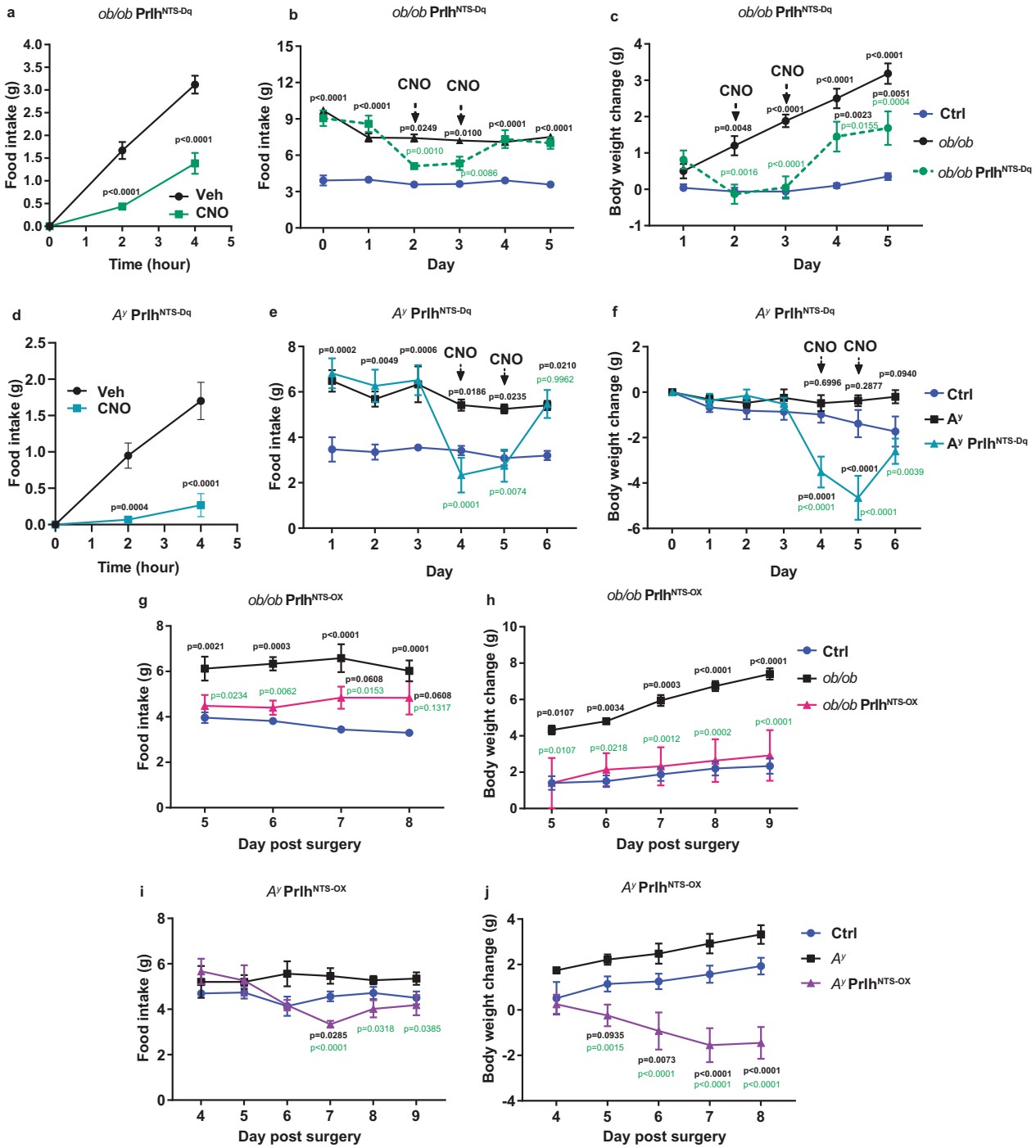

**Fig. 6 PrlhNTS neurons and NTS *Prlh* block feeding and body weight gain in *ob/ob* and *Aʸ* mice. A–F** Acute food intake (**A**, **D**; *n* = 6 animals/group) and daily food intake (**B**, *n* = 7 Ctrl animals, *n* = 6 *ob/ob*;PrlhNTS-Dq animals, and *n* = 5 *ob/ob* animals, **E**, *n* = 6 animals/group) and body weight change (**C**, *n* = 5 Ctrl animals, *n* = 6 animals per group for *ob/ob* and *ob/ob*;PrlhNTS-Dq, **F**, *n* = 6 animals/group; compared to baseline days 0–1) for control (Ctrl) and PrlhNTS-Dq mice on the *Lepob/ob* (*ob/ob*) or *Aʸ* background treated with vehicle for two to three days, CNO (IP, 1 mg/kg; treatment days indicated in panels) for two days, and vehicle for another one or two days. Lean control mice are also included. **G–J** Daily food intake (**G**, *n* = 5 animals per group, **I**, *n* = 7 Ctrl animals, *n* = 6 *Aʸ* animals, and *n* = 5 *Aʸ*;PrlhNTS animals) and body weight change relative to day 0 (**H**, *n* = 5 Ctrl animals, *n* = 6 animals/group for *ob/ob* and *ob/ob*;PrlhNTS-OX, **J**, *n* = 7 Ctrl animals, *n* = 6 *Aʸ* animals and *n* = 5 *Aʸ*;PrlhNTS-OX animals) for lean control (Ctrl) mice, *ob/ob* and *ob/ob*;PrlhNTS-OX mice (**G**, **H**) and *Aʸ* and *Aʸ*;PlrhNTS-OX mice (**I**, **J**). Shown is mean +/− SEM. Two-way ANOVA, sidak's multiple comparisons test was used. Significant or near significant *p* values for comparisons between control mice and *ob/ob* or *Aʸ* groups shown in black; those for comparisons between *ob/ob* and *ob/ob* plus intervention or *Aʸ* and *Aʸ* plus intervention groups shown in green. All experiments were repeated in two independent cohorts of animals with similar results; cohorts were combined for publication.

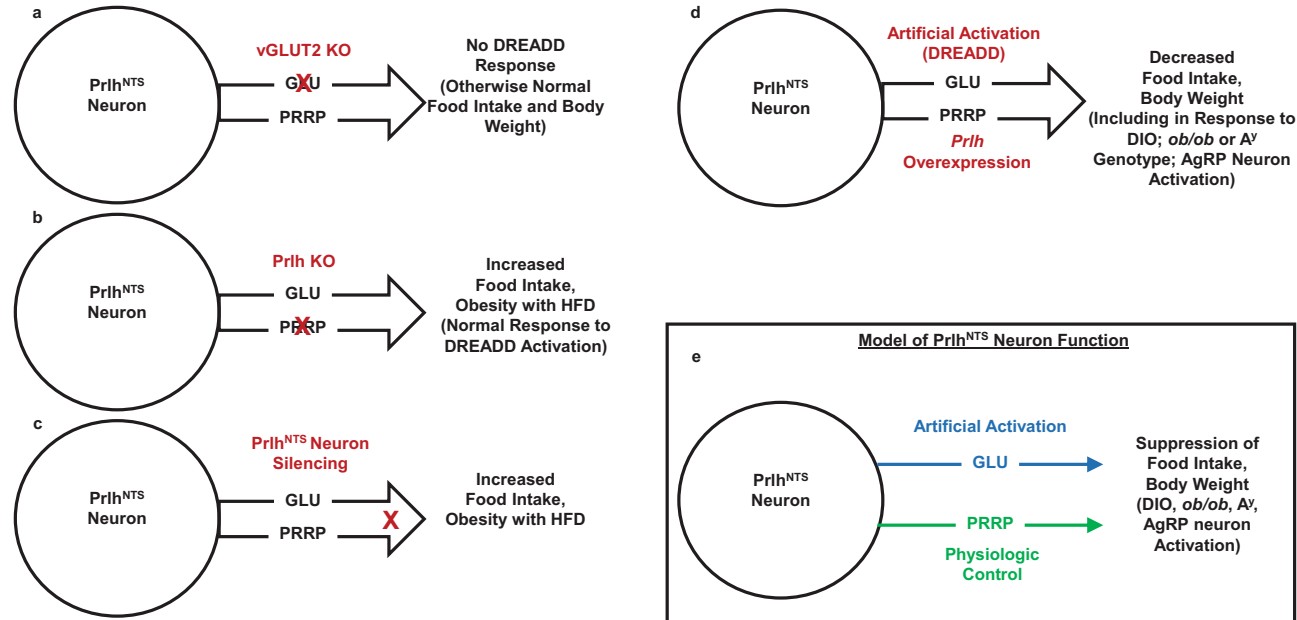

**Fig. 7 Summary and model.** Prlh<sup>NTS</sup> neurons utilize Glutamate (GLU) and PRRP as neurotransmitters. Ablation of vGLUT2 to abrogate GLU signaling blocks the suppression of food intake during DREADD-mediated activation of Prlh<sup>NTS</sup> cells, but does not alter energy balance (**A**). In contrast, ablation of *Prlh* in the NTS does not alter the response to DREADD-mediated activation, but promotes obesity on HFD (**B**). Silencing Prlh<sup>NTS</sup> neurons increases food intake and promotes obesity during HFD exposure (**C**). Increased signaling by Prlh<sup>NTS</sup> cells, either via DREADD-mediated activation or *Prlh* overexpression decreases food intake and body weight in response to multiple obesogenic perturbations, including HFD feeding, AgRP neuron activation, or *ob/ob* or A<sup>y</sup> genotype (**D**). Summary model is shown in (**E**).

determine roles in the control of food intake for PRRP-responsive receptors, PRLHR and NPFFR2, and the neurons that express these receptors in regions innervated by Prlh<sup>NTS</sup> cells. These receptors and the neurons that express them may represent useful targets to non-aversively suppress food intake and body weight for the treatment of obesity and associated conditions.

## Methods

**Animals.** Mice were bred in our colony in the Unit for Laboratory Animal Medicine at the University of Michigan; these mice and the procedures performed were approved by the University of Michigan Committee on the Use and Care of Animals and in accordance with Association for the Assessment and Approval of Laboratory Animal Care and National Institutes of Health guidelines. Mice were provided with food and water *ad libitum* (except as noted below) in temperature-controlled rooms on a 12-h light-dark cycle. For all studies, animals were processed in the order of their ear tag number, which was randomly assigned at the time of tailing (before genotyping). Mice were made DIO by weaning onto HFD (D012492, Research Diets) and subsequent HFD feeding for at least 8 weeks, unless otherwise specified.

We purchased male and female C57BL/6 mice for experiments and breeding from Jackson Laboratories. *Lepr<sup>cre</sup>*, *Calcr<sup>cre</sup>*, and *Rosa26<sup>eGFP-L10a</sup>* mice have been described previously[4,28–30] and were propagated by intercrossing homozygous mice of the same genotype. *Agrp<sup>cre</sup>* mice[31] (Jax stock No.: 012899), *vGlut2<sup>flox32</sup>* (Jax stock No.: 012898), *vGat<sup>cre33</sup>* (Jax stock No.: 016962), *vGlut2<sup>cre33</sup>* (Jax stock No.: 016963), *Lep<sup>ob/+</sup>* (Jax stock No.: 000632), and *A<sup>y/+</sup>* (Jax stock No.: 002468) mice for breeding were purchased from Jackson Labs (Bar Harbor, ME). *Calca<sup>cre-GFP</sup>* (CGRP<sup>GFP</sup>) mice[20] were the generous gift of Richard Palmiter (University of Washington, Seattle, WA). Sequences of all genotyping primers and targeting constructs are shown in Supplementary Table 1.

To generate *Prlh<sup>cre</sup>* mice, we worked with the Molecular Genetics (MG) Core of the Michigan Diabetes Research Center (MDRC). They identified Cas9-dependent sgRNAs upstream and downstream of the *Prlh* STOP codon and injected these, plus an editing template containing homologous genomic sequences along with the internal ribosome entry site (IRES) sequence linked to Cre recombinase followed for the STOP codon within the homologous sequences, into fertilized mouse embryos. The fertilized embryos were implanted into pseudopregnant dams. Resultant pups were screened for the presence of sequences encoding cre recombinase in the context of the *Prlh* genomic sequences; these sequences were amplified and subjected to DNA sequencing. Positive animals were bred to C57Bl6/J mice and the resultant pups were rescreened and resequenced prior to propagation. Subsequent animals were screened for *Prlh<sup>cre</sup>* by PCR using oligos 5'-3': ggtacacgggtcgtgggatc, tgatgtggtttttggggaacaggaa and tcacattgccaaaagacggc.

To generate *Prlh<sup>Flox</sup>* mice, we worked with the MG Core of the MDRC. They identified Cas9-dependent sgRNAs upstream and downstream of *Prlh* exon 2 and subjected these, together with an editing template containing sequences homologous to the region containing the CRISPR/Cas9 cut sites, and with LoxP sites inserted into the Cas9 cut sites, to injection into fertilized embryos, which were implanted into pseudopregnant dams. Resultant pups were screened for the presence of LoxP sites in the context of the relevant *Prlh* genomic sequences; these sequences were amplified and subjected to DNA sequencing. Positive animals were bred to C57Bl6/J mice and the resultant pups were rescreened and resequenced prior to propagation. Subsequent animals were screened for *Prlh<sup>Flox</sup>* by PCR using oligos 5'-3': ctttgagaggaaactctggccac and tgatgtccaccaggtgtagtct.

**TRAP-seq analysis.** Calcr<sup>eGFP-L10a</sup> mice as shown in Fig. 1 were produced by crossing *Calcr<sup>cre</sup>* mice onto the cre-inducible *Rosa26<sup>eGFP-L10a</sup>* background to produce *Calcr<sup>cre/+</sup>;Rosa26<sup>eGFP-L10a/+</sup>* mice, which were then intercrossed to generated double homozygous *Calcr<sup>cre/cre</sup>;Rosa<sup>eGFP-L10a/eGFP-L10a</sup>* mice, which were intercrossed to generate additional Calcr<sup>eGFP-L10a</sup> mice for study. This line was used for one sample of TRAP-seq. Additional Calcr<sup>eGFP-L10a</sup> mice for TRAP-seq were produced by injecting the cre-inducible AAV<sup>Flex-eGFP-L10a</sup> virus into the NTS of *Calcr<sup>cre</sup>* mice; these animals were used for a second sample. For each TRAP sample, material was derived from the dissected dorsal vagal complex of 10–25 individual mice.

We employed anti-eGFP TRAP (as described[29,34]) on hindbrain material from Calcr<sup>eGFP-L10a</sup> mice. Recovered RNA was assessed for quality using the TapeStation (Agilent, Santa Clara, CA). Samples with RNA integrity numbers of 7.5 or greater were prepared using the Illumina TruSeq mRNA Sample Prep v2 kit (catalog nos. RS-122-2001 and RS-122-2002; San Diego, CA), where 0.1–3 μg of total RNA was converted to mRNA using polyA purification. The mRNA was fragmented via chemical fragmentation and copied into first-strand complementary DNA (cDNA) using reverse transcription and random primers. The 3' ends of the cDNA were adenylated, and 6-nucleotide–barcoded adapters were ligated. The products were purified and enriched by PCR to create the final cDNA library. Final libraries were checked for quality and quantity by TapeStation (Agilent) and quantitative PCR using a Library Quantification Kit for Illumina Sequencing platforms (catalog no. KK4835; Kapa Biosystems, Wilmington, MA). They were clustered on the cBot (Illumina) and sequenced four samples per lane on a 50-cycle single-end run on a HiSeq 2000 (Illumina) using version 2 reagents according to the manufacturer's protocols.

52- or 66-base pair single-end reads underwent quality control analysis and filtering using fastp 0.21.0 before alignment to mouse genome build GRCm39 containing custom chromosomes with Cre and eGFP-L10a using STAR 2.7.7a. Enrichment of genes in bead was determined using DESeq2 1.30.1 using paired bead and supernatant samples.

**Viral reagents and stereotaxic injections**. We generated AAV$^{Flex-Prlh}$ by connecting an Integrated DNA Technologies (Coralville, IA)-generated full length cDNA of Prlh with 5′ Nhel and 3′ AscI sticky ends into the NheI/AscI-digested vector from the AAV$^{Flex-hM4Di}$ plasmid[35]. The AAV$^{Flex-Prlh}$ vector, along with that for AAV$^{Flex-TVA + G}$ and the defective pseudotyped rabies-mCherry[16,36] were produced by the University of Michigan viral vector core. AdV-iN-Syn-mCherry was as described previously[37]. AAV$^{Flex-hM3Dq}$[35], AAV$^{Flex-hM4Di}$[35], AAV$^{Flex-TetTox-GFP}$[38], and AAV$^{Flex-ChR2}$ were prepared by the University of North Carolina Vector Core (Chapel Hill, NC). The AAVs used in the manuscript were all serotype AAV8 (except AAV$^{Flex-ChR2}$, which was AAV5).

For injection, following the induction of isoflurane anesthesia and placement in a stereotaxic frame, the skulls of adult mice were exposed. After the reference was determined, a guide cannula with a pipette injector was lowered into the injection coordinates (NTS: A/P, −0.2; M/L, ±0.2; D/V, −0.2 from the obex; ARC: A/P: 1.4 mm, M/L: ±0.25 mm, D/V: 6.0 mm relative to bregma) and 100 nL of virus was injected for each site using a picospritzer at a rate of 5–30 nL/min with pulses. Five minutes following injection, to allow for adequate dispersal and absorption of the virus, the injector was removed from the animal; the incision site was closed and glued. The mice received prophylactic analgesics before and after surgery.

The mice injected with AdV-iN-Syn-mCherry were allowed one week to recover before being euthanized; the mice injected with AAV$^{Flex-hM3Dq}$, AAV$^{Flex-hM4Di}$, AAV$^{Flex-ChR2}$, AAV$^{Cre-mCherry}$, AAV$^{Flex-TetTox-GFP}$, or control viruses were allowed at least 1 week to recover from surgery before experimentation other than the measurement of baseline food intake and body weight. Because AAV$^{Flex-Prlh}$ virus was not tagged with a fluorophore, we sometimes co-injected it with a control GFP- or tdTomato-expressing AAV to aid in injection site validation; we examined PRRP-IR for all animals injected with AAV$^{Flex-Prlh}$.

**Optogenetics**. As described above, AAV$^{Flex-ChR2}$ was injected unilaterally into the ARC of Prlh$^{cre}$; Agrp$^{cre}$ mice to generate Prlh$^{NTSDq}$;AgRP$^{ChR2}$ mice, after which one fiber-optic cannulae (Doric Lenses and Thorlabs) was implanted above the ARC (A/P: 1.4 mm, M/L: ±0.25 mm, D/V: 6.0 mm relative to bregma) and affixed to the skull using Metabond (Fisher). After 3 weeks recovery from surgery, the mice were subjected to optical stimulation using 473 nm wavelength by 1 s of 20 Hz photo stimulation (pulse duration: 10 ms) and 3 s resting with multiple repetitions for up to 2 h, during which time food intake was measured.

**Phenotypic studies**. Prlh$^{NTS-TetTox}$ mice and their controls were monitored from the time of surgery for chow feeding. For stimulation studies, DREADD-expressing mice and their controls that were at least three weeks post-surgery were treated with saline or drugs (CNO, 4936, Tocris) at the onset of dark cycle, and food intake was monitored over 4 h. For chronic food intake and body weight changes, mice were given saline for one to three days prior to injecting saline or CNO (IP, 1 mg/kg at 6 PM) followed by providing CNO (3.33 μg/ml) or vehicle in drinking water for the duration of treatment, followed by saline injections for another one or three days to assess recovery from the treatment. For AAV$^{Flex-Prlh}$ studies to the obese mice, 6 week-old lean control, obese ob/ob and A$^y$ mice with control virus and AAV$^{Flex-Prlh}$ injected in NTS were studied three days post viral injection for up to 2 weeks. For chemogenetic studies in obese mice, 6 week-old lean control, obese ob/ob or obese A$^{y/+}$ mice were injected with control or activating DREADD virus, and were studied two weeks post viral injection. For the Prlh$^{NTS-OX}$ and chemogenetic studies with lean mice, 8 week-old Prlh$^{cre}$; Agrp$^{cre}$ mice with AAV$^{Flex-hM3Dq}$ injection in ARC were verified for food intake in daytime by CNO injection, then AAV$^{Flex-Prlh}$ was delivered in NTS of the validated mice, the food intake study was performed on the 7th day post AAV$^{Flex-Prlh}$ injection. Studies of metabolic rate were performed by placing the animals in chambers of a TSE PhenoMaster apparatus (TSE Systems, Germany) 5 days after surgery. For the optogenetic studies to lean Prlh$^{cre}$; AgRP$^{cre}$ mice, 8 weeks old Prlh$^{cre}$; Agrp$^{cre}$ mice with AAV$^{Flex-hM3Dq}$ injection in NTS and AAV$^{Flex-ChR2}$ in ARC were studied 3 weeks post viral injection.

**Perfusion and immunohistochemistry**. Mice were anesthetized with a lethal dose of pentobarbital and transcardially perfused with phosphate-buffered saline (PBS) followed by 10% buffered formalin. Brains were removed, placed in 10% buffered formalin overnight, and dehydrated in 30% sucrose for 1 week. With use of a freezing microtome (Leica, Buffalo Grove, IL), brains were cut into 30 μm sections. Sections were treated sequentially with 1% hydrogen peroxide/0.5% sodium hydroxide, 0.3% glycine, 0.03% sodium dodecyl sulfate, and blocking solution (PBS with 0.1% triton, 3% normal donkey serum). The sections were incubated overnight at room temperature in rabbit anti-FOS primary antibodies (FOS, #2250, Cell Signaling Technology, 1:1000; GFP, GFP1020, Aves Laboratories, 1:1000; dsRed, 632496, Takara, 1:1000; prolactin-releasing peptide (PRRP), H-008-52, Phoenix Pharmaceuticals, 1:500); antibodies were reacted with species-specific Alexa Fluor-488, -568 or -647 conjugated secondary antibodies (Invitrogen, Thermo Fisher, 1:200). Images were collected on an Olympus (Center Valley, PA) BX53F microscope. Images were pseudocolored using Photoshop 2020 software (Adobe) or Image J (NIH). FOS was quantified by counting cells on one side of the brain of one section that contained anatomically comparable areas of the main region in question from each experimental animal.

**Conditioned taste aversion (CTA)**. Following an overnight fast, chow-fed mice were provided HFD (D012492, Research Diets) paired with the desired stimuli (e.g., LiCl (126 mg/kg, 203637, Sigma), or CNO (1 mg/kg, 4936, Tocris Bioscience)) for 30 mins, followed by an extra hour of access to HFD. On the post-conditioning day, fasted mice received access to both HFD and chow and the consumption of each was measured.

**Statistics**. Data are reported as mean ± standard error of the mean. Statistical analyses of physiologic data were performed with Prism software (version 8). Two-way ANOVA, paired or unpaired $t$ tests were used as indicated in the text and figure legends. $p < 0.05$ was considered statistically significant.

**Reporting summary**. Further information on research design is available in the Nature Research Reporting Summary linked to this article.

## Data availability

All raw data will be made available upon request; data for all graphs is supplied in the supplementary information/source data file. TRAP-seq data are available at GEO: GSE176202. Source data are provided with this paper.

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

## Acknowledgements

We thank B. Roth and K. Deisseroth for AAV plasmid constructs. We thank members of the Myers and Olson labs for helpful discussions, and Qing Zhu from the Molecular Genetics Core for mouse line production. Research support was provided by NIH Award DK104999 (DPO and MGM), and from the Michigan Diabetes Research Center (NIH P30 DK020572, including the Molecular Genetics and Animal Studies Cores), the American Diabetes Association (1-16-PDF-021 to WC), the Marilyn H. Vincent Foundation (MGM), and MedImmune/AstraZenica (to MGM).

## Author contributions

W.C., E.N., J.N.M., W.P. and A.C.R. researched and analyzed data. W.C., E.N., D.P.O., C.J.R., and MGM designed experiments and wrote the paper. All authors reviewed and edited the paper. MGM is the guarantor of the paper.

## Competing interests

CJR is an employee of and holds stock in AstraZeneca. DPO and MGM, Jr. receive research funding from AstraZeneca and Novo Nordisk. The other authors declare that they have no conflicts of interest relevant to this paper.
