## [Peer Review File · Nature Communications]

NTS Prlh overcomes orexigenic stimuli and ameliorates dietary and genetic forms of obesityREVIEWER COMMENTS

Reviewer #1 (Remarks to the Author):

Cheng and coworkers present an extensive study of the effects of NTS-derived PRRP on food intake in lean and DIO mice, and the specific interaction with NPY/AgRP neuron driven increase in food intake, including its role in leptin-deficient mice. This manuscript makes significant advances in our understanding of the circuitry and physiology underlying these effects. It also dissects the effects of Prlh neurons in terms of the respective roles for PRRP and the 'co-transmitter' glutamate. This is a beautiful study and my comments are mainly concerned with the presentation, which could be improved to make it more easily accessible for the reader, and with the addition of some information that should be possible to be extracted from the existing data in order to provide stronger evidence still that all conclusions drawn from the experiments are valid. These points are listed in order of appearance in the manuscript rather than absolute importance.

1) Can the authors please provide evidence that the enriched ribosomes indeed originate from Calcr cells, and the ribosomes from the supernatant do not.

2) Please quantify the colocalization of PRRP-IR with Calcr in the different nuclei. What percentage of PRRP neurons express Calcr, and vice versa, and how do absolute numbers compare between NTS and IRT? (Fig 1). Fig. 1 title only refers to NTS, this should probably be rectified

3) Please quantify colocalization between PRRP-IR and GFP-IR in the Prlh-cre mouse for NTS and IRT (and AP?). (Fig 2). The title for Fig 2 should acknowledge that this is under HFD.

4) Fig 2 F and G miss the data for the vehicle treatment described in the legend. Also, for clarity it should be indicated that there were two CNO injections daily, as described in the methods.

5) Fig. 3: A. Labelling is not intuitive. Is the control PrlhNTS-mCherry? Wouldn't that be a better label?

6) Fig 3C: It seems the significance is driven by a change of the intake trajectory of the controls, rather than an effect of the TetTox. Can the authors comment on this? The y-axis should read 'Cumulative Food Intake'.

7) Fig 3E: There is a discrepancy between the figure labelling and the legend. Presumably the x-axis should read 'Weeks under HFD'?

8) Knockout of Prlh in Calcr cells (Fig 4) shows a much milder effect on food intake and bodyweight than silencing of Prlh cells. In order to judge whether this is due to the importance of PRRP vs co-transmitters such as glutamate, or due to incomplete overlap of populations, quantification of the targeted populations would be helpful. This comes back to the quantification in Fig 1 and quantification in Fig 4 B,C. Fig 4D,F show cumulative food intake. Comparing weight gain under chow and under HFD shows a rather small difference for control animals (Fig 4E,G). Is the DiO phenotype very mild in these mice and only appears when Prlh is knocked down?

9) Fig 5B would be better placed in the supplementary figures, as it simply shows the Cre-dependent targeting of the Prlh overexpressing construct, and the reader might be distracted by the fact that this control was performed on leptin receptor expressing cells.

10) Could Figs 5 D-I be made more consistent between food intake and body weight? Presumably each pair of panels is from the same cohort of animals. Therefore, could they be put on matching x-axes? Additionally, Fig 5 I is confusing. The legend says HFD was introduced at time of surgery, whilst the figure indicates that mice were put on HFD one week after surgery.

11) The images in Fig 6 look like the AAV was injected unilateral. However the methods state that injections were done bilaterally. Which is correct?

12) Fig 7: The title alludes to the effects of PRRP analogues, However, the data is not shown. Can this please be added.

13) Supplementary Fig 6 would benefit from quantification of cFos activation, particularly in regards to the competition between physiological cFos activation, CNO-induced cFos and its suppression by overexpression of PRRP from the NTS.

14) Please be consistent with abbreviation for adenovirus (AdV or Ad)

15) Page 6: rabies-mediated single-synapse retrograde tracing. Also the use of 'afferent' could be seen as misleading, because the standard definition for afferent is 'towards the CNS' rather than towards specific neurons.

16) Suppression of food intake is very strong and lasting, and strongly differential for DIO. What could be the reason? It is reassuring to see a negative CTA, but some disruption seems likely. LiCl is a very strong adverse stimulus. I am somewhat circumspect of whether the design of this CTA study would reveal a slightly less potent adverse stimulus than LiCl. Consequently, in the absence

of any additional independent evidence, I would be more cautious about the conclusion that the reduction in food intake and body weight seen in this study is not related to adverse effects, but purely to suppression of appetite.

17) Page 8: crossing floxed Prlh with Calcr-Cre leads to lack of Prlh in all Calcr cells, not only in the NTS.

18) What was the pulse duration in the optogenetic studies (page 22)?

19) Which experiments were sCT and exendin 4 used in (Page 22)

20) Which FOS AB was used in which studies? The methods gives two different protocols. Maybe a table of all ABs and viruses used would be helpful?

21) Methods for CTA (page 24): what were the doses for LiCl? Peptides? Why 0.8mg/kg CNO, when 1mg/kg was used in other experiments? In fact the legend for Supplementary Fig 2 states 1mg/kg CNO was used. Which figure is correct?

22) The discussion claims that amplifying signals from Prlh NTS neurons attenuates food intake and obesity in melanocortin-deficient mice (page 14). Where does this come from? No experiments were performed on such mice. If so, the results need to be included in the results section and figures.

23) The discussion on brainstem satiety signals (page 14-15) mentions GLP-1. The appropriate references there would be to gcg/ppg neurons rather than GLP-1Rs, particularly given the recent paper by Brierley et al demonstrating the separation of brain and peripheral GLP-1.

24) The authors suggest on page 15 that PRRP signaling by PrlhNTS cells may play a more important role in preventing the hedonic overconsumption of calories than in the inhibition of caloric intake appropriate to weight maintenance in lean mice. Given that activation of the Prlh cells in DIO animals almost completely wipes out food intake, do they think that is likely. Also, normal chow consumption is still strongly suppressed. I would argue that such a statement needs more targeted experimental evidence.

25) This study has provided a large number of rather interesting results. Can I suggest to include a summary schematic that puts all these findings together into one comprehensive diagram for the reader.

26) On page 16 the authors suggest that the Prlh neurons do not activate the PBN CGRP neurons. This is a very interesting point, but the authors have no evidence to directly support this statement. They solely base this on the absence of a positive CTA response (though as addressed in my point 16 I do not feel that this is compelling as it stands). Can the authors provide staining with anti-CGRP to demonstrate that their cFos signal in PBN does not colocalise with CGRP neurons. This would also provide support for the authors' final statement: These receptors and the neurons that express them may represent useful targets to non-aversively suppress food intake and body weight for the treatment of obesity and associated conditions. This statement is not justified in the absence of more supporting evidence as suggested above.

Reviewer #2 (Remarks to the Author):

This paper presents a systematic investigation of how PRLH neurons in the NTS influence food intake and energy balance. The authors show that activation/silencing of NTS-PRLH neurons inhibits/increases food intake, with particularly striking effects in animals on a high-fat diet. Knockout of the PRLH gene from CalcR cells increases body weight on HFD. Overexpression of PRLH in NTS-PRLH neurons reduces food intake, again with enhanced effects in mice on a HFD. Finally, PRLH activation/overexpression is shown to counteract several orexigenic signals arising from the forebrain (AgRP activation, leptin deficiency, agouti).

This is an excellent paper. The experiments are logically designed, the effect sizes are large, and the message is clear. The effects of PRLH stimulation in some obesity models is striking. I have only minor suggestions for improvement.

1. The basic finding in Figure 1 (overlap between CalcR and PRLH neurons) has already been published several times. Space in the main figures would be better used to present the novel rabies and projections data (currently in Supplemental Figure 1) and the novel feeding data (Sup. Fig. 2).

2. It would helpful to more extensively document the rabies/projection findings and preferably quantify. Do the images in Supp. Fig. 1b/c really show the only regions detected or just the most abundant? Some of the images are hard to evaluate.

3. The title gives the initial impression that this is a paper about AgRP neurons, but the findings are broader than that, and a more general title might be better.

Zachary Knight

REVIEWER COMMENTS

Reviewer #1 (Remarks to the Author):

Cheng and coworkers present an extensive study of the effects of NTS-derived PRRP on food intake in lean and DIO mice, and the specific interaction with NPY/AgRP neuron driven increase in food intake, including its role in leptin-deficient mice. This manuscript makes significant advances in our understanding of the circuitry and physiology underlying these effects. It also dissects the effects of Prlh neurons in terms of the respective roles for PRRP and the 'co-transmitter' glutamate. This is a beautiful study and my comments are mainly concerned with the presentation, which could be improved to make it more easily accessible for the reader, and with the addition of some information that should be possible to be extracted from the existing data in order to provide stronger evidence still that all conclusions drawn from the experiments are valid. These points are listed in order of appearance in the manuscript rather than absolute importance.

Response: We thank the referee for these supportive comments and for the thorough review of our manuscript. We have modified the manuscript (as detailed below) in response to the comments; we believe that this process has strengthened the manuscript further.

1) Can the authors please provide evidence that the enriched ribosomes indeed originate from Calcr cells, and the ribosomes from the supernatant do not.

Response: We have now provided the fold enrichment for *Calcr* (2.8-fold), *Gfp* (2.7-fold), and *Cre* (4.9-fold) in the TRAP material compared to supernatant. Note that these and all enrichment values are artificially lowered by the non-quantitative recombination mediated by *Calcr^{cre}* on the *Rosa26^{eGFP-L10a}* reporter background. This information is now provided in the results section, on page 5.

2) Please quantify the colocalization of PRRP-IR with Calcr in the different nuclei. What percentage of PRRP neurons express Calcr, and vice versa, and how do absolute numbers compare between NTS and IRT? (Fig 1). Fig. 1 title only refers to NTS, this should probably be rectified

Response: 61% of PRLH-IR neurons colocalize with *Calcr^{cre}/GFP* in the NTS; 62% of the few neurons in the IRT colocalize. Note that colocalization percentages likely represent underestimates due to the non-quantitative recombination mediated by *Calcr^{cre}* on the *Rosa26^{eGFP-L10a}* reporter background; indeed, quantification of *Calcr^{cre}*-mediated PRRP-IR ablation in the NTS of *Prlh^{Calcr}KO* mice suggest that 85-90% of NTS PRRP-IR neurons express *Calcr^{cre}*. These findings are reported on pages 5 and 8 of the results. We have amended the figure title (now Supplementary figure 1).

3) Please quantify colocalization between PRRP-IR and GFP-IR in the *Prlh-cre* mouse for NTS and IRT (and AP?). (Fig 2). The title for Fig 2 should acknowledge that this is under HFD.

Response: 100% of PRRP-IR cells in the NTS contain *Prlh^{cre}/GFP*, while 74% of *Prlh^{cre}/GFP* neurons contain PRRP-IR. The presence of *Prlh^{cre}/GFP* cells that do not

contain PRRP-IR may results from a failure of PRRP-IR detection in some *Prlh^{cre}* cells. These findings are included in the text of the results (page 6).

We have amended the title to the Figure (now Figure 1) to note that the data are from both lean and DIO mice in the revised figure.

4) Fig 2 F and G miss the data for the vehicle treatment described in the legend. Also, for clarity it should be indicated that there were two CNO injections daily, as described in the methods.

Response: These animals were given a single injection of CNO as a loading dose and then provided with CNO in their drinking water (3.33 ug/ml) for 2 days. We have now clarified this in the methods and in the figure legend.

5) Fig. 3: A. Labelling is not intuitive. Is the control PrlhNTS-mCherry? Wouldn't that be a better label?

Response: Thank you for pointing this out. We have amended the label; it is now Figure 2C.

6) Fig 3C: It seems the significance is driven by a change of the intake trajectory of the controls, rather than an effect of the TetTox. Can the authors comment on this? The y-axis should read 'Cumulative Food Intake'.

Response: We have now made note of this in the main text (page 8). We have corrected the y-axis labels for the panels (now Figures 2E and 2G).

7) Fig 3E: There is a discrepancy between the figure labelling and the legend. Presumably the x-axis should read 'Weeks under HFD'?

Response: Good point- thank you. We have changed the x-axis label of panels 2E and 2G in the revised figure to "Weeks on Chow" and "Weeks on HFD," respectively.

8) Knockout of Prlh in *Calcr* cells (Fig 4) shows a much milder effect on food intake and bodyweight than silencing of Prlh cells. In order to judge whether this is due to the importance of PRRP vs co-transmitters such as glutamate, or due to incomplete overlap of populations, quantification of the targeted populations would be helpful. This comes back to the quantification in Fig 1 and quantification in Fig 4 B,C. Fig 4D,F show cumulative food intake. Comparing weight gain under chow and under HFD shows a rather small difference for control animals (Fig 4E,G). Is the DiO phenotype very mild in these mice and only appears when Prlh is knocked down?

Response: We have now added *Calcr^{cre}* (GFP)/PRRP-IR colocalization data (see main text, page 8) and have added PRRP-IR cell counts to the figure (Figure 3D). These data demonstrate that detection of PRRP-IR cell bodies is 85-90% decreased in *Prlh^{Calcr}* KO mice compared to controls. We agree that it is possible that GLU signaling may contribute to the physiologic control of food intake and energy balance in the absence of PRRP signaling in these mice, and have noted this possibility in the text of the discussion (page 16-17). Note that these mice are on a segregating background and are thus only modestly sensitive to DIO at baseline; we have now noted this on page 8 of the text.

9) Fig 5B would be better placed in the supplementary figures, as it simply shows the Cre-dependent targeting of the Prlh overexpressing construct, and the reader might be distracted by the fact that this control was performed on leptin receptor expressing cells.

Response: Thank you for the suggestion; we have moved the panel (now Supplemental Figure 5B).

10) Could Figs 5 D-I be made more consistent between food intake and body weight? Presumably each pair of panels is from the same cohort of animals. Therefore, could they be put on matching x-axes? Additionally, Fig 5 I is confusing. The legend says HFD was introduced at time of surgery, whilst the figure indicates that mice were put on HFD one week after surgery.

Response: We have attempted to improve the consistency of labeling across panels (now Figures 4E-H), but note that there is no day 0 measurement for food intake, as these measurements began following surgery. Also, parameters were measured at different time intervals for some experiments. We have also clarified that HFD in former Figure 5I (now Figure 4H) began one week after surgery in the figure legend. Note also that we have added new data showing energy expenditure for lean, chow-fed mice in metabolic cages (Figure 4D).

11) The images in Fig 6 look like the AAV was injected unilateral. However the methods state that injections were done bilaterally. Which is correct?

Response: The reviewer is correct: the AAV-ChR2 was injected unilaterally (now figure 5). Thank you for catching this. We have corrected the methods accordingly.

12) Fig 7: The title alludes to the effects of PRRP analogues, However, the data is not shown. Can this please be added.

Response: Unfortunately, the analogs that we had employed proved to be unspecific, forcing us to omit the data. We have amended the figure title (now figure 6) to reflect the lack of PRRP analog data.

13) Supplementary Fig 6 would benefit from quantification of cFos activation, particularly in regards to the competition between physiological cFos activation, CNO-induced cFos and its suppression by overexpression of PRRP from the NTS.

Response: We have quantified the data (Supplementary Figure 6G, K).

14) Please be consistent with abbreviation for adenovirus (AdV or Ad)

Response: We have now used AdV throughout.

15) Page 6: rabies-mediated single-synapse retrograde tracing. Also the use of 'afferent' could be seen as misleading, because the standard definition for afferent is 'towards the CNS' rather than towards specific neurons.

Response: We have amended the text and omitted the potentially confusing usage of "afferent."

16) Suppression of food intake is very strong and lasting, and strongly differential for

DIO. What could be the reason? It is reassuring to see a negative CTA, but some disruption seems likely. LiCl is a very strong adverse stimulus. I am somewhat circumspect of whether the design of this CTA study would reveal a slightly less potent adverse stimulus than LiCl. Consequently, in the absence of any additional independent evidence, I would be more cautious about the conclusion that the reduction in food intake and body weight seen in this study is not related to adverse effects, but purely to suppression of appetite.

Response: We agree that Prlh^{NTS} neurons promote a strong suppression of food intake. Regarding the potential for a small, difficult-to-detect aversive response: We have previously shown a similar lack of CTA (and a small reinforcing effect) for Calcr^{NTS} neuron activation (Cheng, et al., Cell Metabolism, 2020). Furthermore, the Prlh^{NTS} CTA data actually show a trend toward reinforcement, if anything, suggesting that these cells are unlikely to mediate aversive responses. We cannot prove a negative, of course, and have thus softened the wording around this issue (see especially page 16).

17) Page 8: crossing floxed Prlh with Calcr-Cre leads to lack of Prlh in all Calcr cells, not only in the NTS.

Response: Agreed, thanks. Now corrected.

18) What was the pulse duration in the optogenetic studies (page 22)?

Response: The stimulation pattern was 1 second of 20Hz photo stimulation with 473 nm wavelength and 3 second resting for up to 2 hours, so the pulse duration is 0.05 second, we now have updated this to the optogenetics session in the Materials and Methods (page 23).

19) Which experiments were sCT and exendin 4 used in (Page 22)

Response: These were used as controls for the studies in which we attempted to utilize PRRP analogs (which turned out to be non-specific and were thus removed from the manuscript, see point 12, above). We have removed this from the methods.

20) Which FOS AB was used in which studies? The methods gives two different protocols. Maybe a table of all ABs and viruses used would be helpful?

Response: We have updated the methods section to reflect our use of a single FOS-directed antibody throughout the manuscript, and have removed information about a different FOS-directed antibody that was not used in the present manuscript.

21) Methods for CTA (page 24): what were the doses for LiCl? Peptides? Why 0.8mg/kg CNO, when 1mg/kg was used in other experiments? In fact the legend for Supplementary Fig 2 states 1mg/kg CNO was used. Which figure is correct?

Response: The LiCl dose was 126 mg/kg, and the CNO dose was 1 mg/kg throughout the studies in the present manuscript; we have corrected the text throughout. We have removed the reference to peptide studies that were not included with the manuscript (see points 12 and 19, above).

22) The discussion claims that amplifying signals from Prlh NTS neurons attenuates food intake and obesity in melanocortin-deficient mice (page 14). Where does this come

from? No experiments were performed on such mice. If so, the results need to be included in the results section and figures.

Response: Our apologies- we intended to refer to mice that were deficient in melanocortin signaling, because ASP overexpression in A^y mice blocks signaling by melanocortin receptors. We have now corrected the text.

23) The discussion on brainstem satiety signals (page 14-15) mentions GLP-1. The appropriate references there would be to gcg/ppg neurons rather than GLP-1Rs, particularly given the recent paper by Brierley et al demonstrating the separation of brain and peripheral GLP-1.

Response: We have now cited the Brierley paper and amended the text.

24) The authors suggest on page 15 that PRRP signaling by PrlhNTS cells may play a more important role in preventing the hedonic overconsumption of calories than in the inhibition of caloric intake appropriate to weight maintenance in lean mice. Given that activation of the Prlh cells in DIO animals almost completely wipes out food intake, do they think that is likely. Also, normal chow consumption is still strongly suppressed. I would argue that such a statement needs more targeted experimental evidence.

Response: Thank you for this suggestion, we have now reworded the text.

25) This study has provided a large number of rather interesting results. Can I suggest to include a summary schematic that puts all these findings together into one comprehensive diagram for the reader.

Response: Thank you for this suggestion- we have now added a schematic (Figure 7).

26) On page 16 the authors suggest that the PrIH neurons do not activate the PBN CGRP neurons. This is a very interesting point, but the authors have no evidence to directly support this statement. They solely base this on the absence of a positive CTA response (though as addressed in my point 16 I do not feel that this is compelling as it stands). Can the authors provide staining with anti-CGRP to demonstrate that their cFos signal in PBN does not colocalise with CGRP neurons. This would also provide support for the authors' final statement: These receptors and the neurons that express them may represent useful targets to non-aversively suppress food intake and body weight for the treatment of obesity and associated conditions. This statement is not justified in the absence of more supporting evidence as suggested above.

Response: Thank you for this suggestion. We have now added the requested data (Figure 1L-N (and quantification in the text, page 7)).

Reviewer #2 (Remarks to the Author):

This paper presents a systematic investigation of how PRLH neurons in the NTS influence food intake and energy balance. The authors show that activation/silencing of NTS-PRLH neurons inhibits/increases food intake, with particularly striking effects in animals on a high-fat diet. Knockout of the PRLH gene from CalcR cells increases body weight on HFD. Overexpression of PRLH in NTS-PRLH neurons reduces food intake,

again with enhanced effects in mice on a HFD. Finally, PRLH activation/overexpression is shown to counteract several orexigenic signals arising from the forebrain (AgRP activation, leptin deficiency, agouti).

This is an excellent paper. The experiments are logically designed, the effect sizes are large, and the message is clear. The effects of PRLH stimulation in some obesity models is striking. I have only minor suggestions for improvement.

Response: We thank the referee for these supportive comments. We have modified the manuscript (as detailed below) in response to the comments; we believe that this process has further strengthened the manuscript.

1. The basic finding in Figure 1 (overlap between CalcR and PRLH neurons) has already been published several times. Space in the main figures would be better used to present the novel rabies and projections data (currently in Supplemental Figure 1) and the novel feeding data (Sup. Fig. 2).

Response: Thank you for this suggestion. We have now moved the former Figure 1 into the supplement. We have also moved the feeding data (from the former Supplemental Figure 2) into the main figures (as part of the new Figure 1). Because the tracing data are not as key as the functional data we present, we have left these in the supplement, however (now Supplemental Figure 2).

2. It would helpful to more extensively document the rabies/projection findings and preferably quantify. Do the images in Supp. Fig. 1b/c really show the only regions detected or just the most abundant? Some of the images are hard to evaluate.

Response: The data show all areas with significant (more than 1-2 scattered cells or fibers) tracing. We have now provided images taken at lower exposure times to facilitate readability. We have added new summary diagrams regarding the relative strength of innervation to each tracing study (Supplementary figure 2Bg and 2Cg).

3. The title gives the initial impression that this is a paper about AgRP neurons, but the findings are broader than that, and a more general title might be better.

Response: Thank you for this suggestion; we have now revised the title, as suggested.

REVIEWERS' COMMENTS

Reviewer #1 (Remarks to the Author):

The authors have satisfactorily addressed all of my concerns raised. There are just three small points that require more clarification.

1) The authors have added the following details about their optogenetic stimulation: 1s at 20Hz with 0.05s pulse duration.

However, 20Hz means 1 pulse every 50ms. Does this mean light was on continuously for 1s?

2) The authors added quantification of cFOS induced in CGRP neurons. Can they please provide data on whether the difference between control ($2 \pm 0.3\%$) and CNO stimulation of PrLh-NTS neurons ($9 \pm 4\%$) was significant or not.

3) The authors have added cFOS quantification, but have not added in the methods how this was done. Is this the average across a specific number of sections analysed, or is it the sum from all sections analysed, or how was it done?

Reviewer #2 (Remarks to the Author):

The authors have addressed my comments, this is a great paper and ready for publication.

Response to Reviews:

Reviewer #1 (Remarks to the Author):

The authors have satisfactorily addressed all of my concerns raised. There are just three small points that require more clarification.

Response: We thank the reviewer for the supportive comments.

1) The authors have added the following details about their optogenetic stimulation: 1s at 20Hz with 0.05s pulse duration.

However, 20Hz means 1 pulse every 50ms. Does this mean light was on continuously for 1s?

Response: We thank the reviewer for catching this misunderstanding. The light was on for 10 ms of each 50 ms cycle.

2) The authors added quantification of cFOS induced in CGRP neurons. Can they please provide data on whether the difference between control ($2 \pm 0.3\%$) and CNO stimulation of Prh-NTS neurons ($9 \pm 4\%$) was significant or not.

Response: We have now provided these data in the text of the results (they were not significantly different) (page 7).

3) The authors have added cFOS quantification, but have not added in the methods how this was done. Is this the average across a specific number of sections analysed, or is it the sum from all sections analysed, or how was it done?

Response: We have now added this information to the methods (page 25)- we quantified one anatomically comparable section per mouse (from within the main region being studied).

Reviewer #2 (Remarks to the Author):

The authors have addressed my comments, this is a great paper and ready for publication.

Response: We thank the reviewer for their support.